# UniChart: A Universal Vision-language Pretrained Model for Chart Comprehension and Reasoning

**Ahmed Masry**[♣,*], **Parsa Kavehzadeh**[♣,*], **Xuan Long Do**[♠,†], **Enamul Hoque**[♣], **Shafiq Joty**[♠♦]

[♣]York University, Canada

[♠]Nanyang Technological University, Singapore, [♦]Salesforce AI

{parsaka, enamulh}@yorku.ca

srjoty@ntu.edu.sg

ahmed.elmasry24653@gmail.com, xuanlong.do@u.nus.edu

## Abstract

Charts are widely used for data analysis, providing visual representations and insights into complex data. To facilitate chart-based data analysis using natural language, several downstream tasks have been introduced recently such as chart question answering and chart summarization. However, existing methods for these tasks often rely on pretraining on language or vision-language tasks, neglecting the explicit modeling of chart structures (e.g., how chart elements are related to each other). To address this, we first build a large corpus of charts covering diverse topics and visual styles. We then present UniChart, a pretrained model for chart comprehension and reasoning. UniChart encodes the relevant text, data, and visual elements of charts and then uses a chart-grounded text decoder for text generation. We propose several chart-specific pretraining tasks that include: (i) low-level tasks to extract the visual elements (e.g., bars, lines) and data from charts, and (ii) high-level tasks to acquire chart understanding and reasoning skills. Our experiments demonstrate that pretraining UniChart on a large corpus with chart-specific objectives, followed by fine-tuning, yields state-of-the-art performance on four downstream tasks. Moreover, our model exhibits superior generalizability to unseen chart corpus, surpassing previous approaches that lack chart-specific objectives and utilize limited chart resources.

## 1 Introduction

Information visualizations such as bar charts and line charts are commonly used for analyzing data, inferring key insights and making informed decisions (Hoque et al., 2022). However, understanding important patterns and trends from charts and answering complex questions about them can be cognitively taxing. Thus, to facilitate users in analyzing charts, several downstream NLP tasks over

charts have been proposed recently, including chart question answering (Masry et al., 2022; Kantharaj et al., 2022; Lee et al., 2022), natural language generation for visualizations (Obeid and Hoque, 2020; Shankar et al., 2022) and automatic data story generation (Shi et al., 2020).

A dominant strategy to tackle these downstream tasks is to utilize pretrained models (Su et al., 2020; Li et al., 2020b; Kim et al., 2021; Cho et al., 2021) trained on langauge and vision tasks (Du et al., 2022). However, although effective, such models may not be optimal for chart-specific tasks because they are trained on large text corpus and/or image-text pairs without any specific focus on chart comprehension. In reality, charts differ from natural images in that they visually communicate the *data* using *graphical marks* (e.g., bars, lines) and *text* (e.g., titles, labels, legends). Readers can discover important patterns, trends, and outliers from such visual representation (Munzner, 2014). Existing pretrained models do not consider such unique structures and communicative goals of charts. For instance, Pix2Struct (Lee et al., 2022) is a pretrained image-to-text model designed for situated language understanding. Its pretraining objective focuses on screenshot parsing based on HTML codes of webpages, with a primary emphasis on layout understanding rather than reasoning over the visual elements. MatCha (Liu et al., 2022b) extends Pix2Struct by incorporating math reasoning and chart data extraction tasks, but it still lacks training objectives for text generation from charts and it was trained on a limited number of charts.

In this work, we present UniChart, a pretrained model designed specifically for chart comprehension and reasoning. UniChart is pretrained on a large corpus of charts and it aims to serve as a Universal model for various chart-related downstream tasks (Fig. 1). Inspired by the model architecture from Kim et al. (2022), UniChart consists of two modules: *(1)* a chart encoder, which takes the chart

---

[*]Equal contribution.

[†] Now affiliated with NUS.

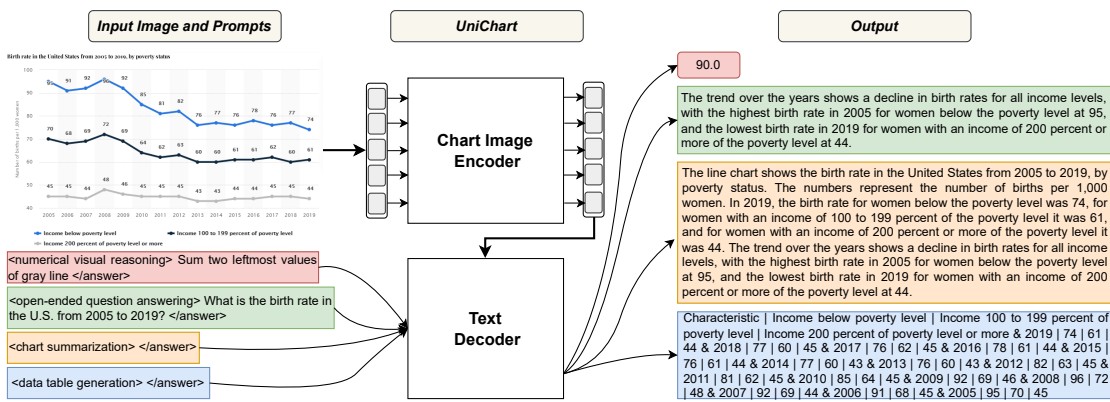

Figure 1: Our UniChart model with different pretraining objectives. The model consists of two main modules: Chart Image Encoder, and Text Decoder. Four different pretraining objectives are specified in different colors; *data table generation*, *chart summarization*, *numerical and visual reasoning*, and *open-ended question answering*.

image as input, and *(2)* a text decoder, trained to decode the expected output based on the encoded image and the text input fed in the decoder as task prompt. We performed pretraining on a diverse set of 611K charts that we collected from multiple real-world sources. Our pretraining objectives include both low-level tasks focused on extracting visual elements and data from chart images, as well as high-level tasks, intended to align more closely with downstream applications. One key challenge for pretraining was that most charts in the corpus do not come with informative summaries, which are critical for various downstream tasks. To address this challenge, we used knowledge distillation techniques to leverage large language models (LLMs) for opportunistically collecting chart summaries, which were then used during pretraining.

We conducted extensive experiments and analysis on various chart-specific downstream tasks to evaluate the effectiveness of our approach. Specifically, we evaluated UniChart on two chart question answering datasets, ChartQA (Masry et al., 2022) and OpenCQA (Kantharaj et al., 2022), and found that it outperformed the state-of-the-art models in both cases. For chart summarization, UniChart achieves superior performance in both human and automatic evaluation measures such as BLEU (Post, 2018) and ratings from ChatGPT (OpenAI, 2022). Moreover, UniChart achieved state-of-the-art results in the Chart-to-Table downstream task. Finally, our model showed improved time and memory efficiency compared to the previous state-of-the-art model, MatCha, being more than 11 times faster with 28% fewer parameters.

Our primary contributions are: (i) A pretrained model for chart comprehension with unique low-level and high-level pretraining objectives specific

to charts; (ii) a large-scale chart corpus for pretraining, covering a diverse range of visual styles and topics; (iii) extensive automatic and human evaluations that demonstrate the state-of-the-art performance of UniChart across various chart-specific benchmark task while optimizing time and memory efficiency. We have made our code and chart corpus publicly available at https://github.com/vis-nlp/UniChart.

## 2 Related Work

### 2.1 Vision-language Pretraining

Pretrained models have dominated in many vision and language tasks (Du et al., 2022). Building a pretrained vision-language model typically involves three steps. First, textual input is usually encoded using BERT-based encoder (Lu et al., 2019; Radford et al., 2021; Li et al., 2021, 2022). Second, for the input image, some prior studies utilize Fast-RCNN (Ren et al., 2015) to encode the sequence of object regions as the image features (Li et al., 2019; Lu et al., 2019; Chen et al., 2020). However, this method may neglect some crucial regions in an image. Recent approaches favor encoding the image as a whole (Huang et al., 2020, 2021; Li et al., 2021, 2022) by using ResNet (He et al., 2016) or ViT (Dosovitskiy et al., 2021). Third, to fuse the textual and visual features, prior work mostly either designs a fusion encoder (Tan and Bansal, 2019; Su et al., 2020; Cho et al., 2021; Kim et al., 2021) or a dual encoder (Radford et al., 2021; Jia et al., 2021; Li et al., 2022). Finally, multiple common cross-modal pretraining tasks have been designed such as image-text matching (Chen et al., 2020; Li et al., 2020a), cross-modal contrastive learning (Radford et al., 2021; Jia et al., 2021) and generation tasks such as visual question answering (Cho et al., 2021; Wang et al., 2021).

Our work is also related to multimodal document understanding tasks that involve analyzing the textual content, layout, and visual elements of documents (Xu et al., 2020b,a; Wang et al., 2022; Huang et al., 2022; Kim et al., 2022; Tang et al., 2022). These tasks can be addressed using encoder-only and encoder-decoder architectures. Encoder-only models rely on OCR engines to extract text from document images and use BERT-like encoders augmented with specialized embeddings to encode layout and visual features (Xu et al., 2020b,a; Wang et al., 2022; Huang et al., 2022). In contrast, encoder-decoder architectures combine transformer-based encoders with autoregressive text decoders for text generation tasks related to documents (Tang et al., 2022; Kim et al., 2022; Lee et al., 2022). While Tang et al. (2022) incorporates an OCR tool to supplement the vision encoder, Kim et al. (2022) and Lee et al. (2022) operate in an end-to-end manner without external OCR engines. In line with the latter approach, our model adopts an end-to-end encoder-decoder architecture (Kim et al., 2022).

In general, the above work focuses on training on large image-text pairs or text corpus, lacking focus on chart understanding. One exception is MatCha (Liu et al., 2022b), a pretrained chart model based on Pix2Struct (Lee et al., 2022), which achieved SoTA on chart question answering and summarization tasks. However, MatCha's pretraining tasks mainly focus on data table generation without focusing on text generation tasks. The model is also pretrained with reasoning tasks using the textual datasets which might limit its visual reasoning ability. Our model is trained on a larger corpus with chart-specific pretraining objectives, including visual reasoning and text generation, making it more versatile for various chart-related tasks.

## 2.2 Chart-related Downstream Tasks

There has been growing interest in solving various chart-related tasks. Chart question answering (ChartQA) tackles questions about charts, with benchmarks like (Methani et al., 2020) and (Masry et al., 2022) targeting factoid questions involving visual and arithmetic reasoning. Open-ended question answering (OpenCQA) task requires an explanatory answer by reasoning with the chart content (Kantharaj et al., 2022). Finally, Chart-to-Text generates natural language summaries from input charts (Shankar et al., 2022), while Chart-to-Table generates underlying data tables (Choi et al.,

2019). We evaluate our model on these four chart-related tasks, as they involve the interaction between language and vision and have publicly available datasets. There are a few other tasks such as infographics understanding (Mathew et al., 2022) and question answering with science diagram (Kembhavi et al., 2016), however, in this work, we only focus on chart-related tasks.

## 3 Chart Pretraining Corpus

To build a large and diverse corpus with various styles, topics, and storage formats, we crawled charts from various online sources. Additionally, we utilized publicly available chart datasets suitable for pretraining. The collected charts can be categorized into two types: charts with underlying data tables and charts without data tables.

### 3.1 Charts with Data Tables

Charts with an underlying data table are collected in three ways: (*i*) utilize existing datasets, (*ii*) extract SVG charts, and (*iii*) data augmentation.

• **Utilize Existing Datasets** Our goal was to train the model based on real-world data, thus, we did not consider the ones that are generated from synthetic data (Kafle et al., 2018; Kahou et al., 2018). In particular, we used the following five chart datasets for which the underlying data tables were available: (*i*) Statista (statista.com) (Shankar et al., 2022), (*ii*) Our World In Data or OWID (ourworldindata.org) (Masry et al., 2022), (*iii*) Organisation for Economic Co-operation and Development or OECD (oecd.org) (Masry et al., 2022), (*iv*) PlotQA (Methani et al., 2020), and (*v*) a subset of the ChartInfo (ChartInfo, 2022) dataset that provides bounding box annotations for data encoding marks (e.g., bars in a bar chart).

• **Extract SVG Charts:** We extracted charts in SVG format from the Chartblocks and Plotly datasets of the Beagle corpus (Battle et al., 2018). These charts do not come with data tables, but the data can be extracted accurately from the SVG elements. The steps for preparing these charts are: (1) identify axis labels and legends using specific class names of HTML attribute, (2) extract bounding boxes of chart elements (e.g., bars, line) using SVG attribute properties (e.g., `size` and `location` of `<rect>`), (3) construct the underlying data table by iterating through each of the `<g>` elements to find data values of each data attribute. When data labels are absent, we utilize the scale information based on the axis labels and tick marks of the chart

and the bounding box information of data encoding marks to recover the data values.

• **Data Augmentation** We further augmented the corpus by creating charts from publicly available data tables. We used the The Web Data Commons (WDC) (WDC, 2022), which used Common Crawl[1] to collect a large amount of structured data. The charts are created in the following steps:

(i) *Data pre-processing:* Since many tables in WDC contain more than three columns, we decomposed so that tables are suitable for creating desired chart types (e.g., bars, lines, and pie charts). In particular, we automatically analyze the data type of each column (e.g, numeric vs. categorical) and then randomly choose one column with numeric data values and one/two column(s) with categorical data. We also limit the maximum number of rows of the table to 8 so that the corresponding chart can fit within reasonable screen space.

(ii) *Chart generation*: To generate visually diverse charts, we used the D3 (Bostock et al., 2011) library that provides great flexibility in terms of creating diverse visualization styles. We also employed Vega-Lite (Satyanarayan et al., 2016) which creates charts based on declarative JSON syntax. We used simple heuristics for determining chart types from the data table (Mackinlay et al., 2007). We created four types of charts: (1) vertical simple bar charts with one numeric data column, (2) vertical grouped bar charts, (3) pie charts, and (4) line charts (both single series and multi-series).

(iii) *Visual diversification*: To create visually diverse charts resembling real-world variations, we manipulated the following visual style properties: (1) **Colors and shapes**: Color schemes from Color-Brewer[2] and Tableau[3] were chosen for categorical data attributes. We also varied shape properties such as bar thickness, line types (e.g., continuous vs dotted), and legend shape types (e.g., rect, circle). (2) **Position and distance**: We also varied bar positions and distances with respect to axis labels. (3) **Guides**: Charts may contain additional guides such as grids, so we generate charts with and without grids to diversify styles.

Fig. 2 depicts a visually diverse set of charts created using this augmentation process. In total, we created a total of 189,839 charts (Table 4).

---

[1]https://commoncrawl.org/

[2]https://colorbrewer2.org/

[3]tableau.com

## 3.2 Charts without Data Tables

Many online charts are available only as images, without corresponding data tables. However, they can still be valuable for large-scale pretraining as we can extract chart elements and rich textual contents (e.g., titles, surrounding texts, captions) using object detection and optical character recognition (OCR) techniques. We collected image chart datasets such as LineCap (Mahinpei et al., 2022) and Neural Caption Generation (Spreafico and Carenini, 2020) since they provide high-quality summaries. We also used the Pew dataset from (Shankar et al., 2022) and further augmented it by an crawling additional 1K charts. Finally, we used the ExcelChart400K dataset (Luo et al., 2021) which only provides bounding boxes without underlying data tables. We also considered other existing image chart datasets such as Vis30K (Chen et al., 2021) and VisImage (Deng et al., 2020), but they are not suitable as they usually have poor resolution and lack meaningful textual content (e.g., titles).

## 3.3 Augmentation by Knowledge Distillation for Chart-to-text Generation Tasks

Chart-related downstream tasks such as chart summarization (Shankar et al., 2022) and open-ended question answering (Kantharaj et al., 2022) require generating informative and relevant texts. However, for most of the charts in the pretraining corpus, there are either no associated summaries or the summaries that are collected opportunistically such as the Statista dataset (Shankar et al., 2022) lack quality (e.g., too short and not very informative). Training on such substandard "ground-truth" summaries can negatively affect the overall model performance as shown in text summarization (Kryscinski et al., 2019; Clark et al., 2021). Indeed, Goyal et al. (2022) and Liu et al. (2023b) have recently shown that human raters prefer summaries generated by LLMs, especially the ones that are instruction-tuned such as InstructGPT (Ouyang et al., 2022), compared to the reference summaries in various text summarization datasets. Consequently, the instruction-tuned LLMs have been successfully used as a annotator in several recent studies (DING et al., 2023; Qin et al., 2023).

Inspired by these findings, we leveraged Instruct-GPT to generate coherent and relevant text. Specifically, we prompted `text-davinci-003` by providing the underlying data table as input and one exemplar (i.e., 1-shot in-context learning). Since

generating summaries for thousands of charts by calling OpenAI API is quite costly, we devised a knowledge distillation approach. We first used `text-davinci-003` to create a small dataset of 3700 summaries for different chart types. Next, we finetuned Flan-T5 XL (Chung et al., 2022) on this dataset. Finally, we utilized the finetuned Flan-T5 model to generate summaries for charts that do not have an associated summary. More details about this approach can be found in Appendix A.2.

## 3.4 Datasets Analysis

Our chart pretraining corpus has over 611K charts covering a diverse range of bar charts, line charts, and pie charts (Table 4). Data tables of *Simple* charts have two columns (simple bar charts or single-series line charts), whereas *Complex* charts involve at least three columns (e.g., stacked or group bar charts, line charts with multiple lines). The first two chart groups in Table 4 come with an underlying data table which cover over 80% of the corpus. The bottom group contains five datasets which only provide charts in image format without a data table[4] and cover about 20% of the corpus. Bar charts make up the majority portion (58.51%), followed by line charts (32.94%) and pie charts (9.39%). About 60% of the charts have multiple columns in their data tables, while 40% of the charts have only two columns.[5] The corpus also covers a diverse range of topics including technology, economy, politics, health, and society. To ensure a fair evaluation, we excluded charts found in the validation and test sets of the downstream tasks from our pretraining corpus. Details about the linguistics of the corpus textual elements can be found in Appendix A.3.

## 4 Method

We propose UniChart, a unified pretrained model for chart comprehension and reasoning. This section first introduces the UniChart architecture followed by its pretraining objectives.

### 4.1 Model Architecture

UniChart consists of two main modules: a chart image encoder and a text decoder as shown in Fig. 1.

---

[4]The ExcelChart400K dataset only provides bounding box annotations of chart elements and we used this dataset for data value estimation task during pretraining.

[5]Since we do not have access to the chart types of Pew dataset, we manually tagged random 200 samples from each of these datasets to estimate the chart type distribution.

• **Chart Image Encoder** In order to effectively encode a chart image, an encoder needs to identify and interpret three different types of chart components: (1) textual elements (axis labels and legends), (2) visual elements (e.g., bars, lines), and (3) the layout that arranges textual and visual elements within a chart. Since this has a similarity with document image (e.g., receipts) understanding, our chart image encoder builds upon the encoder of one of the recent state-of-the-art document image understanding models, Donut (Kim et al., 2022).

Donut offers an OCR-free architecture. The model is pretrained using an OCR-pseudo task, where it sequentially generates the encoded text in a document image, following the order from the top-left corner to the bottom-right corner of the image. As a result, we did not have to run an external OCR module like CRAFT (Baek et al., 2019) and Parseq (Bautista and Atienza, 2022), which improved time and memory efficiency throughout our training pipeline. Donut employs Swin Transformer (Liu et al., 2021) architecture as the image encoder. To encode the chart image features, the images are split into non-overlapping patches, which are then processed using shifted window-based multi-headed self-attention and MLP layers to produce the image embeddings.

• **Text Decoder** Similar to Donut (Kim et al., 2022), we use the BART (Lewis et al., 2019) decoder for generating the output. The textual (task-specific) prompts are fed to the decoder and the decoder has to generate the output by conditioning on the prompted context (see Fig. 1).

## 4.2 Pretraining Objectives

Our pretraining objectives include low-level tasks that are more focused on retrieving the underlying data from the chart images and high-level tasks that align closely with the downstream tasks.

• **Data Table Generation** A chart creates a visual representation of a data table by mapping each data attribute (e.g., 'country', 'population') to corresponding visual attributes (e.g., `x-positions`, `height`) of graphical marks (e.g, bars). An effective chart comprehension and reasoning model should be able to deconstruct the structured underlying data table by recovering such mappings. To this end, we propose the data table generation task in which we ask the model to generate the flattened data table given a chart image.

A vast amount of charts available online are

| Dataset | Data Table Generation | Numerical & Visual Reasoning | Open-ended Question Answering | Chart Summarization |
|---|---|---|---|---|
| Pew | 0 | 0 | 5,295 | 5,295 |
| Statista, OECD, OWID | 144,147 | 679,420 | 126,009 | 126,009 |
| PlotQA | 155,082 | 2,414,359 | 157,070 | 157,070 |
| LineCap | 0 | 0 | 2,821 | 2,821 |
| Neural Caption | 0 | 0 | 100 | 306 |
| Beagle | 3,972 | 51 | 0 | 0 |
| ChartInfo | 1,796 | 21,949 | 0 | 0 |
| Data Aug. | 189,792 | 2,218,468 | 189,802 | 189,802 |
| ExcelChart | 106,897 | 0 | 0 | 0 |
| Total | **601,686** | **5,334,247** | **481,097** | **481,303** |

Table 1: Number of examples for each task in pretraining.

stored as bitmap images without access to the underlying data. It is important to learn how to recover data values when the chart data is not available. Therefore, we also introduce the data value estimation task, in which the model is asked to generate the scale of the graphical marks (e.g., bars, line points) as a percentage of the chart plot area. We obtain these scales by dividing the bars or line points heights (bounding boxes) by the height of the chart plot area and rounding the result to two decimal places. At the final stage, we use charts for which both data tables and object bounding boxes are available as well as charts for which at least the bounding box annotations are available, e.g., ExcelCharts from (Luo et al., 2021).

• **Numerical & Visual Reasoning** Many downstream applications over charts may involve numerical and visual reasoning with the chart elements such as chart QA and summarization. For example, the model may need to apply a series of mathematical and logical operations such as addition, subtraction and comparisons to answer a question. To inject such reasoning skills into the model, we design template-based numerical reasoning tasks where the model is trained to execute/perform the most common mathematical operations over the chart data values. We manually analyzed the existing task datasets (e.g., ChartQA) to find the most common operations (*e.g.,* sum, average, difference, etc.) and constructed 90 templates that we utilize to generate synthetic question and answer pairs. All the templates are provided in Appendix A.8.

• **Open-ended Question Answering** It is very common for users to ask open-ended questions over charts (Kantharaj et al., 2022). Such questions often ask for answers that require high-level reasoning and explanations. To improve the capability of the model in answering open-ended questions, we follow previous work (Shi et al., 2022) to generate synthetic open-ended QA pairs. Specifically, a T5 model (Raffel et al., 2020) pretrained on SQuAD (Rajpurkar et al., 2016) is employed to generate an open-ended question for each summary. The sentence containing the answer in the summary then serves as the answer to its generated question.

• **Chart Summarization** Image captioning is a fundamental problem in AI in which the machines need to summarize the main content of the image in the textual form. This task has been studied extensively (Vinyals et al., 2015; Herdade et al., 2019; Hu et al., 2021; Li et al., 2022). We follow previous work (Vinyals et al., 2015; Xia et al., 2021) to pretrain our model on this task to further enhance the model's capability in generating textual descriptions from the chart image. As discussed in §3.3, we used mostly the summaries generated from GPT models provided by OpenAI either directly or through a knowledge distillation step.

## 4.3 Downstream Tasks

In addition to zero-shot evaluation, we also adapt UniChart by finetuning it on a downstream task. We consider four downstream tasks: *(1) Factoid Chart Question Answering:* we use ChartQA (Masry et al., 2022), which is a benchmark consisting of factoid question-answer pairs for charts with a particular focus on visual and logical reasoning questions; *(2) Complex Chart Question Answering:* we consider OpenCQA (Kantharaj et al., 2022), another QA benchmark in which answers are explanatory descriptions; *(3) Chart Summarization:* we use Chart-to-Text (Shankar et al., 2022), a large-scale benchmark for chart summarization; *(4) Chart-to-Table:* we use ChartQA for both finetuning and evaluation. Moreover, we evaluate the pretrained model in a zero-shot setup on the WebCharts dataset (Choi et al., 2019), a collection of 300 charts obtained from the web.

## 4.4 Experiments Setup

To minimize the computational resource requirements, we initialize our model from the base Donut weights (Kim et al., 2022). Our pretraining process consists of two stages. In the first stage, we set the input image resolution to 512x512 and pretrain for 300K steps. In the second stage, we increase the input image resolution to 960x960 and pretrain for an additional 100K steps. Table 6 shows the hyperparameters we used in pretraining and finetuning our model on each downstream task. All our experiments were carried out using one 4-A100 (40GB), one 4-A100 (80GB), and one 4-V100 (32 GB) GPU machines.

| Model | #Params | ChartQA (RA) | | | OpenCQA (BLEU) | Chart-to-Text (BLEU) | | Chart-to-Table ($RNSS \mid RMS_{F_1}$) | |
|---|---|---|---|---|---|---|---|---|---|
| | | aug. | human | avg. | OpenCQA | Pew | Statista | ChartQA | WebCharts |
| VisionTaPas (Masry et al., 2022) | - | 61.44 | 29.60 | 45.52 | - | - | - | - | - |
| T5 (Masry et al., 2022) | 222M | 56.96 | 25.12 | 41.04 | 9.28 | 10.49 | 35.29 | - | - |
| VL-T5 (Masry et al., 2022) | - | 56.88 | 26.24 | 41.56 | 14.73 | - | - | - | - |
| Pix2Struct (Lee et al., 2022) | 282M | 81.6 | 30.5 | 56.0 | - | 10.3 | 38.0 | - | - |
| MatCha (Liu et al., 2022b) | 282M | **90.2** | 38.2 | 64.2 | - | 12.2 | **39.4** | 85.21 \| 83.49 | 44.37 \| 17.94 |
| UniChart | 201M | 88.56 | **43.92** | **66.24** | **14.88** | **12.48** | 38.21 | **94.01 \| 91.10** | **60.73 \| 43.21** |

Table 2: Evaluation results on four public benchmarks: ChartQA, Chart-to-Text, OpenCQA, and Chart-to-Table. All the results are calculated after finetuning UniChart pretrained checkpoint except for WebCharts (zero-shot).

# 5 Evaluation

## 5.1 Baselines & Evaluation Metrics

We compare our model against five baselines: (1) *T5* (Raffel et al., 2020), a unified seq2seq Transformer model that achieved state-of-the-art (SoTA) results on various text-to-text tasks, including question answering and summarization; (2) *VL-T5* (Cho et al., 2021), a T5-based model that unifies Vision-Language (VL) tasks as text generation conditioned on multimodal inputs and achieved SoTA results on OpenCQA (Kantharaj et al., 2022); (3) *VisionTapas* (Masry et al., 2022), an extension of TaPas (Herzig et al., 2020), a SoTA table encoder, adapted for QA over charts; (4) *Pix2Struct* (Lee et al., 2022), a pretrained image-to-text model for visual language understanding and achieved SoTA results on document understanding tasks; and (5) *MatCha* (Liu et al., 2022b), an adapted version of Pix2Struct for charts that is further pretrained on math reasoning and chart data extraction tasks, achieving SoTA results on Chart-to-Text (Shankar et al., 2022) and ChartQA (Masry et al., 2022).

To evaluate our approach, we follow previous works (Lee et al., 2022; Shankar et al., 2022; Masry et al., 2022; Kantharaj et al., 2022; Liu et al., 2022b) and utilize Relaxed Accuracy (RA) for ChartQA and BLEU (Post, 2018) for text-generation tasks (Chart-to-Text and OpenCQA). However, the BLEU score has limitations as it primarily focuses on n-gram matching between the generated and reference texts, overlooking important factors such as semantic similarity, informativeness, and factual correctness (Goyal et al., 2022). Therefore, we conduct a human evaluation and ChatGPT-driven study to assess and compare these crucial aspects in the outputs of different models (§5.3). Finally, we use Relative Number Set Similarity (RNSS) (Masry et al., 2022) and Relative Mapping Similarity (RMS) (Liu et al., 2022a)

| Summary | Human | ChatGPT | p-value |
|---|---|---|---|
| UniChart ZeroShot | **3.97** | **3.18** | 1.70e-10 |
| UniChart Finetuned | 2.86 | 2.37 | 2.32e-8 |
| MatCha (Liu et al., 2022b) | 2.50 | 2.18 | 0.0020 |
| Gold (Shankar et al., 2022) | 3.19 | 2.73 | 2.13e-6 |

Table 3: Average Informativeness scores from Human and ChatGPT-based evaluation.

metrics to evaluate the Chart-to-Table task.

## 5.2 Main Results

As shown in Table 2, UniChart outperforms previous state-of-the-art models, MatCha and VL-T5, on the ChartQA and Chart-to-Text (Pew) datasets, although it shows slightly lower performance on Chart-to-Text (Statista). The performance gap is more prominent on the challenging human-written questions in the ChartQA benchmark (Masry et al., 2022), where our model's pretraining objectives tailored to visual and numerical reasoning give it a significant advantage. UniChart also achieved a higher BLUE score compared to the SoTA VL-T5 model on OpenCQA benchmark, which demonstrates our model's capability in generating explanatory answers for questions about charts. Finally, UniChart surpasses MatCha's performance on two datasets, demonstrating its generalizability across diverse visual styles, even in a zero-shot setup on unseen charts (WebCharts). Overall, these results establish UniChart as the SoTA model for chart comprehension and reasoning tasks.

To further assess the impact of our different pretraining objectives on our model's performance, we conducted ablation studies. We observe that removing various pertaining objectives led to a slight decrease in performance (Table 8). The decrease in performance is particularly noticeable when the Numerical Reasoning pretaining task is removed, highlighting the importance of this task in imbuing numerical abilities into our model. More details of this experiment can be found in Appendix A.4.

## 5.3 Human and ChatGPT Evaluation

As discussed in §5.1, reference-based metrics like BLEU have relatively low correlations with human judgments (Belz and Reiter, 2006; Tan et al., 2015; Liu et al., 2023a), and generated texts with very high such scores can be of a very poor quality (Smith et al., 2016). Therefore, we decided to conduct a human evaluation to measure the quality of summaries generated by different models. We focus on following criteria in the chart summarization task:*(1) Informativeness*; *(2) Factual Correctness*; and*(3) Semantic Levels that characterize the content of the summary*. More details about the criteria can be found in Appendix A.5.

We randomly picked 150 sample charts from Chart2text Statista test split and asked 3 human annotators to rate four summaries for each chart based on *informativeness* out of 1 to 5. The order of exposure of summaries to the annotator was randomized to avoid any potential bias. Summaries for each chart were rated by one annotator except for the first 100 charts for which we had two annotators to measure the agreement. We computed Krippendorff's alpha (Krippendorff, 2011) to measure inter-annotator agreement and found a moderate level of agreement with an alpha coefficient of 0.54. We further utilize ChatGPT for evaluating the same 150 samples, as LLMs have demonstrated their effectiveness as evaluators for text generation tasks (Luo et al., 2023; Liu et al., 2023a; Gao et al., 2023; Fu et al., 2023). We define the informativeness criteria and rating scheme to ChatGPT and then employ ChatGPT to generate evaluation steps. We then send these evaluation steps along with the data table of the chart and the summary to ChatGPT to obtain ratings (see Appendix A.5 for details).

Table 3 shows the result of human evaluation on chart summarization based on informativeness criteria. We notice that annotators preferred ZeroShot version of our model which generates summaries that are more similar to those generated by GPT, rather than gold summaries. The finetuned version of UniChart was also rated higher compared to SoTA MatCha (Liu et al., 2022b). The finetuned UniChart model also produces fewer factual errors compared to Matcha and the ZeroShot version (Appendix A.5 and Table 7). We observe that the ratings provided by ChatGPT are roughly consistent with the human annotators' scores in terms of informativeness criteria. Moreover, we conducted a statistical test (p-value) for ratings from humans

and ChatGPT, with the null hypothesis that the ratings are two independent samples. The p-values in each row in Table 3 demonstrate that it is very infrequent that two rating samples are independent based on the observed ratings. Also in terms of different semantic contents, the ZeroShot model tends to contain more sentences with high-level visual patterns and trends. A previous study finds that such high-level insights lead to more reader takeaways compared to the text describing low-level visual encodings like axes and colors (Stokes et al., 2022). Overall, the results above suggest that UniChart model's summaries are more informative with high-level insights and factually accurate than the SoTA (MatCha).

## 5.4 Time and Memory Efficiency

UniChart exhibits significant time efficiency compared to MatCha, as shown in Fig. 4. The gap in speed is more evident on tasks that require the generation of long output sequences (e.g., Chart-to-Text). This difference in speed can be attributed to MatCha's use of a long input sequence (4K) with a quadratic increase in complexity while UniChart's vision encoder relies on sliding windows with a local attention mechanism that scales linearly with the input image size. Moreover, UniChart boasts a smaller parameter count (201M) compared to MatCha (282M), further contributing to its efficiency. As a result, UniChart is highly suitable for real-world applications that prioritize fast inference speeds. More details are provided in Appendix A.7.

## 5.5 Error Analysis and Challenges

We conducted a manual analysis of our model's outputs to identify key challenges faced by existing models.

• **Densely populated charts:** Our model struggles with extracting insights from chart images that contain numerous data elements densely packed in a limited area. This is evident in Figure Fig. 9 (Q3) where our model generates a hallucinated summary due to the complexity of the chart. Increasing model parameters and input image resolution could potentially improve performance in these cases.

• **Numerical reasoning:** Despite efforts to incorporate mathematical skills, our model still encounters difficulties with complex arithmetic calculations (Q2 in Fig. 9). Addressing this challenge involves decoupling arithmetic calculations and reasoning steps by employing external program executors that perform the calculations using the equa-

tions generated by our model (Gao et al., 2022).

• **Factual correctness in generated summaries:** Factual correctness still poses a challenge for autoregressive language models (Lin et al., 2022; OpenAI, 2022; Zhao et al., 2023). Although our finetuned UniChart model produced fewer factual errors compared to MatCha (see Table 7), it still generates some incorrect statements (see Q4 in Fig. 9). This issue can be attributed to factual errors in the pretraining captions generated by ChatGPT.

# 6 Conclusion

We present UniChart, a general purpose pretrained model designed for a broad range of chart-related tasks. Our model incorporates chart-specific pretraining tasks and is trained on a large and diverse collection of charts and corresponding summaries collected opportunistically using LLMs. We conducted both human and ChatGPT evaluations to show the superiority of our method. While our model sets the state-of-the-art record on four different downstream tasks and showed improved time and memory efficiency, the evaluation also reveals opportunities for improvement. We believe that our model and pretraining data will be valuable resources for future research and encourage further exploration in this relatively new area.

## Limitations

While UniChart exhibits state-of-the-art performance on several benchmarks, it suffers from several limitations. Despite the remarkable abilities on the ChartQA dataset, the model still struggles to answer questions that involve compositional mathematical operations. Moreover, we have noticed that the model may hallucinate and produce factually incorrect statements on the text generation tasks such as Chart-to-Text and OpenCQA.

Despite the generalizability of our model on unseen chart image styles (WebCharts), there's still a noticeable drop in performance compared to the performance on the tasks on which the model is finetuned (e.g., ChartQA). Hence, there's still a need for better generalizable chart models for the diverse charts on the Web. One direction is to enlarge our pretraining datasets by crawling millions of chart images from the Web. Since most charts on the Web do not provide high-quality captions or the underlying data table, self-supervised pretraining objectives are needed to benefit from these charts.

Due to the limited computing resources, we did not investigate the effect hyperparameter tuning might have on the performance on the different downstream tasks. Also, although we have noticed the convergence of UniChart at the end of the second stage pretraining, we can not confirm whether further pretraining may improve the performance of our model.

## Ethics Statement

During the dataset collection process, we made sure to comply with the terms and conditions of the different websites we used to crawl our data. Statista[6] provide a permissive license to use their publicly available data for scientific purposes. Pew Research Centre [7] also provide a permissive license to use their data with the condition that we attribute it to the Centre. OECD[8] allows the users to download and publish their data as long as they give appropriate credit to the OECD website. For OWID[9], all their data are provided under the Creative Commons BY license which gives the permission for downloading and publication. Web Data Commons[10], which we used in the data augmentation process, allows the usage of their data under the conditions of the Apache License Software which gives the right to download and publish. Finally, all the remaining datasets (PlotQA, Beagle, ChartInfo, ExcelChart400K, LineCap, and Neural Captions) are publicly available datasets which were released in earlier scientific publications.

Due to the generative nature of our models, they may be abused for misinforming the public by generating factually incorrect responses. Moreover, we can not guarantee that our models may not produce texts that may contain hate speech or harmful content.

## Acknowledgement

The authors would like to thank the anonymous reviewers for their helpful comments. This research was supported by the Natural Sciences & Engineering Research Council (NSERC) of Canada and Canada Foundation for Innovation (CFI).

---

[6] https://www.statista.com/getting-started/publishing-statista-content-terms-of-use-and-publication-rights

[7] https://www.pewresearch.org/about/terms-and-conditions/

[8] https://www.oecd.org/termsandconditions/

[9] https://ourworldindata.org/faqs#can-i-use-or-reproduce-your-data

[10] https://webdatacommons.org/

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

## A  Appendices

### A.1  Data Augmentation

During the data augmentation process, we mainly utilized two of the most popular visualization libraries: D3 (Bostock et al., 2011) and Vegalite (Satyanarayan et al., 2016). Moreover, we have introduced a range of visual variations in terms of color scheme, elements dimensions, shapes, background, .etc (see Fig. 2). This makes our generated chart images closely resemble the real-world charts found on the Web.

### A.2  Data Augmentation by Knowledge Distillation

We select the small dataset (3700 charts) from PlotQA and the augmented charts from WDC, since these datasets are accompanied by the underlying data tables which serve as suitable chart representation for LLMs. Also, they cover a wide range of topics which contributes to the diversity in the generated summaries (§3.1). Fig. 3 shows our process for generating summaries for the charts that have underlying data tables using InstructGPT model (Ouyang et al., 2022). The input mainly consists of one demonstration (table-caption pair) followed by the desired chart data table. The output is the generated summary. Using this mechanism, we generated a small dataset of 3,700 samples. We then finetuned Flan-T5 XL (Chung et al., 2022) on this dataset. To our knowledge, Flan-T5 was the SoTA open-sourced instruction-tuned model during the development of our dataset. After finetuning on our task, we (qualitatively) observed similar performance as `text-davinci-003`. At the final step, we used the finetuned Flan-T5 model to generate summaries for all the charts that do not have an associated summary (e.g., PlotQA, augmented charts, OWID and OECD charts). In this process, we added around 470K summaries for charts in our pretraining corpus. Fig. 5 shows some examples generated by the finetuned Flan-T5.

To benefit more from the capability of GPT models in generating high-quality summaries, we further prompt ChatGPT (`gpt-3.5-turbo`) (OpenAI, 2022) to generate summaries for the charts from Statista and Pew Research and put these in our pretraining corpus instead of the original summaries in the Chart-to-Text benchmark (Shankar et al., 2022). In most cases, we found the summaries from ChatGPT to be more elaborate and of better writing style. For the Pew Research Centre

| Datasets | Chart Type | | | | | | Linguistic Statistics | | |
|---|---|---|---|---|---|---|---|---|---|
| | Bar | | Line | | Pie | #Charts | #Vocab | Avg. Character | Avg. Token |
| | Two-Col | Multi-Col | Two-Col | Multi-Col | | | | | |
| Statista | 71.9% | 15.3% | 8.3% | 2% | 2% | **19,143** | 24,392 | 111.37 | 21.88 |
| OWID | 51.9% | 0.0% | 9% | 38.9% | 0.0% | **60,624** | 3,721 | 85.89 | 16.96 |
| OECD | 49.1% | 0.0% | 3.1% | 47.7% | 0.0% | **64,380** | 1,606 | 65.47 | 14.67 |
| PlotQA | 11.2% | 55.6% | 6.7% | 26.2% | 0.0% | **157,070** | 2,230 | 155.32 | 33.21 |
| Beagle | 29.8% | 27.3% | 24.7% | 17.9% | 0.0% | **3,972** | 11,361 | 78.76 | 20.55 |
| ChartInfo | 31.7% | 51.0% | 8.6% | 8.6% | 0.0% | **1,796** | 13,329 | 120.75 | 26.11 |
| Data Augmentation | 13.3% | 49.3% | 11.7% | 11.1% | 14.3% | **189,836** | 117,244 | 85.62 | 21.16 |
| ExcelChart400K | 11.5% | 32.7% | 12.0% | 22.3% | 27.7% | **106,897** | 515,922 | 138.68 | 27.72 |
| PewResearch | 11.4% | 55.5% | 4.4% | 21.9% | 6.5% | **5,295** | 38,165 | 477.33 | 98.08 |
| LineCap | 0.0% | 0.0% | 15.9% | 84.0% | 0.0% | **2,821** | 16,570 | 102.11 | 24.62 |
| Neural Caption | 0.0% | 0.0% | 100% | 0.0% | 0.0% | **100** | 389 | 117.56 | 27.43 |
| ***Total*** | **21.95%** | **36.56%** | **9.23%** | **23.71%** | **9.39%** | 611,934 | 888,522 | 114.70 | **25.01** |

Table 4: Chart type distribution and linguistic statistics of the chart pertaining corpus. The charts in the last group (magenta) do not come with an underlying data table. The charts generated by the data augmentation process are shown in blue.

| Datasets | #Vocab | Avg. Char | Avg. Token | Avg. Sentence |
|---|---|---|---|---|
| Statista | 72,725 | 450.28 | 106.68 | 4.46 |
| OWID | 58,212 | 463.48 | 105.99 | 4.47 |
| OECD | 24,752 | 414.97 | 95.08 | 4.78 |
| PlotQA | 112,394 | 666.09 | 149.84 | 5.20 |
| Data Aug. | 162,239 | 468.41 | 113.46 | 4.49 |
| PewResearch | 13,449 | 604.04 | 133.01 | 4.66 |
| LineCap | 2018 | 110.82 | 26.24 | 1.87 |
| Neural Caption | 1338 | 262.84 | 53.28 | 3.58 |

Table 5: Statistics about the captions of the datasets used in LM pretraining.

charts, the underlying data tables are not provided. However, we have observed that the underlying data values are written on the visual elements in most of these charts. Hence, we decided to use an OCR tool to extract the layout-preserving texts from the chart images, and then feed it into Chat-GPT to generate the summaries as shown in Fig. 6. We realized that ChatGPT is capable of understanding a chart from the OCR data.

### A.3  Dataset Analysis

The linguistic characteristics of the textual elements vary across different datasets, with charts from PlotQA and PewResearch often having longer text elements (e.g., axis labels, legends, titles), while augmented data and Beagle datasets contain shorter text (Table 4, right). In Table 5, we further provide linguistic statistics for the summaries of the datasets used in the summary generation task at pretraining.

### A.4  Ablation study

To further assess the impact of our different pretraining objectives on our model's performance, we conducted ablation studies. Due to computational limitations, we focused on pretraining the model only on the lower image size (512x512) and compared it against the corresponding main model (512x512). From Table 8, we observe that removing the Chart Summarization or Open-ended Question Answering objectives led to a slight decrease in performance on ChartQA. We attributed this to the abundance of numerical reasoning examples in pretraining. However, removing the Numerical Reasoning pretaining task led to a substantial decrease in performance on ChartQA, highlighting the importance of this task in imbuing numerical abilities into our model. Pretraining the model without the Data Table Generation objective resulted in a relatively weak performance in the ChartQA benchmark, underscoring the importance of understanding underlying data tables of charts in answering reasoning questions.

### A.5  Human and ChatGPT Evaluation

As discussed in section §5.3, we evaluate the following three criteria in the human evaluation study: *(1) Informativeness* which measures how much information from the chart the summary covers. Ideally, an informative summary should contain high-level insights from the chart, such as important patterns, trends, and outliers in data; *(2) Factual*

| Experiment | # Epochs/Steps | Learning Rate | Batch Size | GPUs | Saving Mechanism |
|---|---|---|---|---|---|
| **Pretraining** | | | | | |
| First-stage/ablations | 300K steps | 1e-4 | 160 | 4xA100 82GB | each 50K steps |
| Second-stage | 100K steps | 1e-4 | 80 | 4xA100 82GB | each 50K steps |
| **Finetuning (main 960x960 model)** | | | | | |
| ChartQA | 20 epochs | 5e-5 | 24 | 4xV100 32GB | each 1 epoch |
| Chart-to-text Pew | 200 epochs | 5e-5 | 48 | 4xA100 40GB | each 5 epochs |
| Chart-to-text Statista | 100 epochs | 5e-5 | 48 | 4xA100 40GB | each 5 epochs |
| OpenCQA | 200 epochs | 5e-5 | 24 | 4xV100 32GB | each 5 epochs |

Table 6: Training details for pretraining and finetuning experiments.

| Criteria | ZeroShot | Finetuned | MatCha | Gold |
|---|---|---|---|---|
| Factually incorrect sents | 13.45% | 9.63% | 21.97% | **3.59%** |
| Elemental/encoded sents | 19.42% | 26.06% | **29.61%** | 21.07% |
| Statistical/relational sents | **57.41%** | 33.42% | 34.07% | 34.70% |
| Perceptual/cognitive sents | **6.98%** | 1.41% | 0.31% | 5.39% |
| Contextual/domain-specific sents | 1.36% | 14.44% | 7.32% | **20.56%** |

Table 7: Human evaluation on summaries for 150 random samples from Chat2text Statista test split.

| | ChartQA | | |
|---|---|---|---|
| Model | aug. | human | avg. |
| UniChart (512x512) | 85.84 | 43.60 | 64.72 |
| No Chart Summarization | 84.96 | 42.72 | 63.84 |
| No Open-ended Question Answering | 85.52 | 42.96 | 64.24 |
| No Numerical & Visual Reasoning | 84.08 | 35.44 | 59.76 |
| No Data Table Generation | 83.84 | 42.24 | 63.04 |

Table 8: UniChart ablations on ChartQA benchmark.

*Correctness* which considers how accurate the summary is. A factually correct summary should only contain information (e.g. numbers, events, entities) that is true and/or supported by the chart; *(3) Semantic Levels defined by (Lundgard and Satyanarayan, 2021)* which categorize the content of summaries across four levels: visual encoding (e.g., axis, legends, color), statistical/relational (e.g., min, max, avg.), perceptual/cognitive (e.g., describing overall trends, complex patterns, or outliers), and context/domain specific information. Our process for evaluating the informativeness is explained in §5.3. For factual correctness and semantic level measures, the annotator goes through each sentence of the summary to determine whether the sentence contains any factual error and what levels of semantic content are present for that sentence. Table 7 shows the results of our human evaluation study on factual correctness, and different semantic levels.

Fig. 7 shows an overview of the paradigm we use in our ChatGPT-driven evaluation study. Fig. 8 depicts the interface we used in our human evaluation study.

### A.6 Error Analysis

Fig. 9 shows the models performance on challenging samples. Q1 and Q2 examples are two visual numerical reasoning questions about charts which look challenging for SoTA models. Q3 is an example of an overpopulated chart with so many data elements which confuses the model to generate insightful summary. Finally, Q4 shows a factual error in a generated summary from finetuned UniChart.

### A.7 Time and Memory Efficiency

To compare the time efficiency, we measure the average inference time of the models on three benchmarks: ChartQA, Chart-to-Text (Pew), and Chart-to-Text (Statista) using 10 random samples from each benchmark. The experiments were conducted on Google's Colab platform with cpu type. Overall, UniChart shows much faster inference times compared to MatCha as shown in Fig. 4.

### A.8 Templates for Numerical and Visual Reasoning Question Generation

Table 9 is the list of the templates we used to generate numerical and visual reasoning questions.

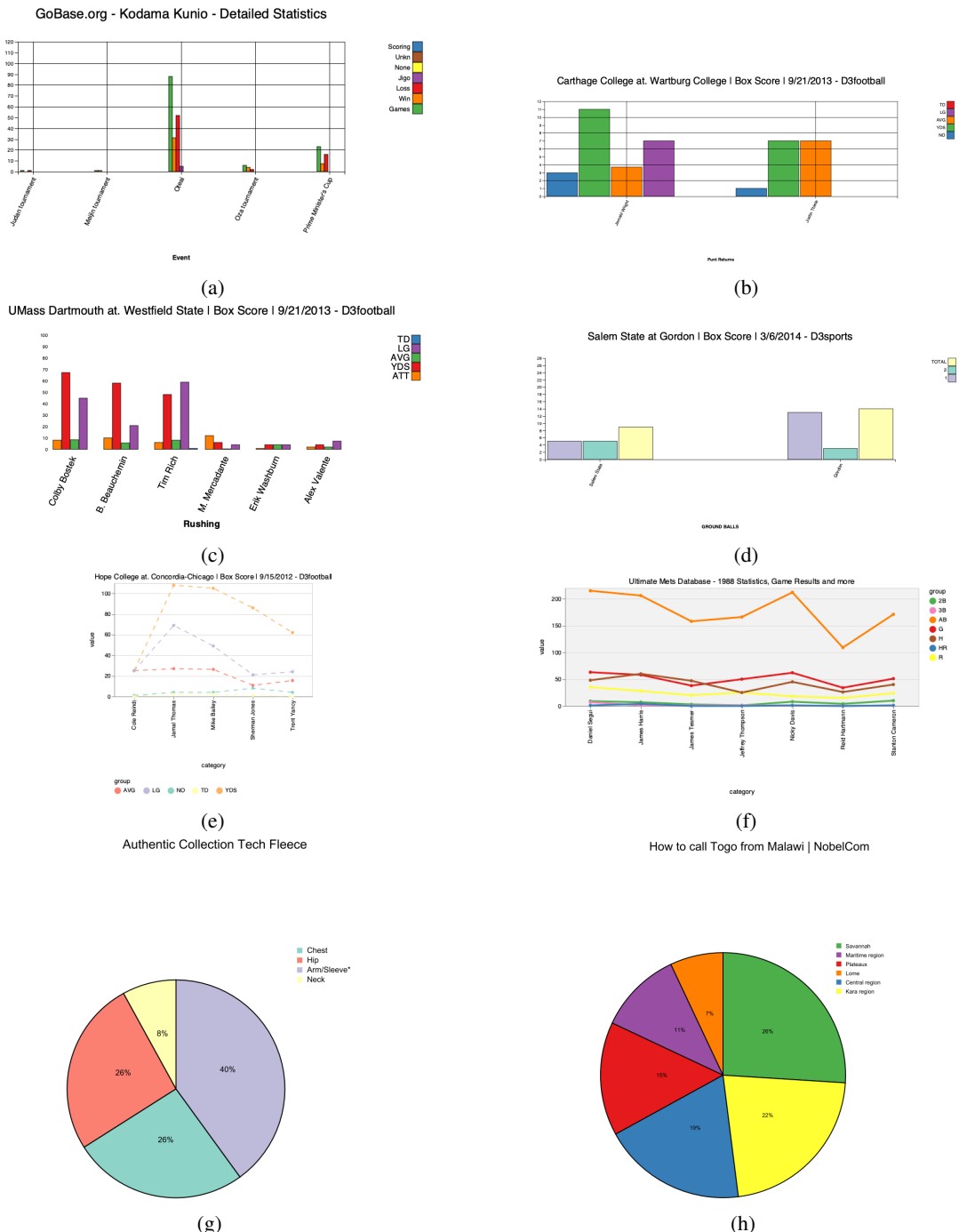

Figure 2: Visually diverse charts generated by D3 and Vegalite for WDC corpus (Fig. 2a, Fig. 2b, Fig. 2c, Fig. 2d, Fig. 2g and Fig. 2h from D3-WDC, Fig. 2e and Fig. 2f from Vegalite-WDC. Visual factors like color scheme, width of bars, and existence of grids and axis labels are different among the samples.

**Analyze the following bar chart in one paragraph and round the numbers. The numbers represent Rating of statistical capacity (0-100).**
**Chart: Years | Botswana | Samoa & 2010 | 30 | 50.0 & 2011 | 30 | 50.0 & 2012 | 40 | 60.0 & 2013 | 30 | 60.0**
**Summary: The bar chart shows the rating of statistical capacity for Botswana and Samoa from 2010 to 2013. In 2010, both countries had a statistical capacity rating of 30, but by 2012, Samoa's rating improved to 60 while Botswana's rating only increased to 40. In 2013, both countries had a rating of 60 for Samoa and 30 for Botswana. It can be concluded that Samoa has shown a better improvement in its statistical capacity compared to Botswana over the four-year period.**

Analyze the following bar chart in one paragraph and round the numbers. The chart title is Percentage of population having access to electricity and the numbers represent Access to resorces (% of population).
Chart: Years | Kiribati | St. Lucia | Venezuela & 1990.0 | 48.86 | 81.14 | 98.0 & 2000.0 | 52.53 | 85.41 | 99.0 & 2010.0 | 55.8 | 87.87 | 100.0 & 2012.0 | 59.33 | 90.88 | 100.0
Summary:

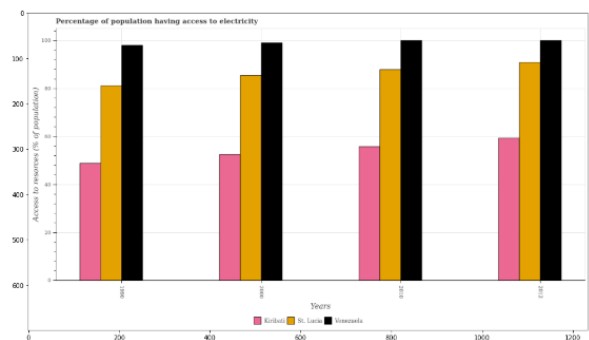

**InstructGPT**

The bar chart shows the percentage of population having access to electricity in Kiribati, St. Lucia, and Venezuela from 1990 to 2012. In 1990, 48.86%, 81.14%, and 98% of the population had access to electricity in Kiribati, St. Lucia, and Venezuela respectively, while in 2012, the percentages increased to 59.33%, 90.88%, and 100% respectively. It is evident that all three countries have experienced a significant increase in access to electricity over the 22-year period, with Venezuela having the highest access rate of 100% in 2012.

Figure 3: An example of the performance of *InstructGPT* in generation summaries for data tables. On the left side, the red text is a full example of a demonstration and its summary followed by the demonstration for the target chart. The paragraph in green shows the summary generated by the model.

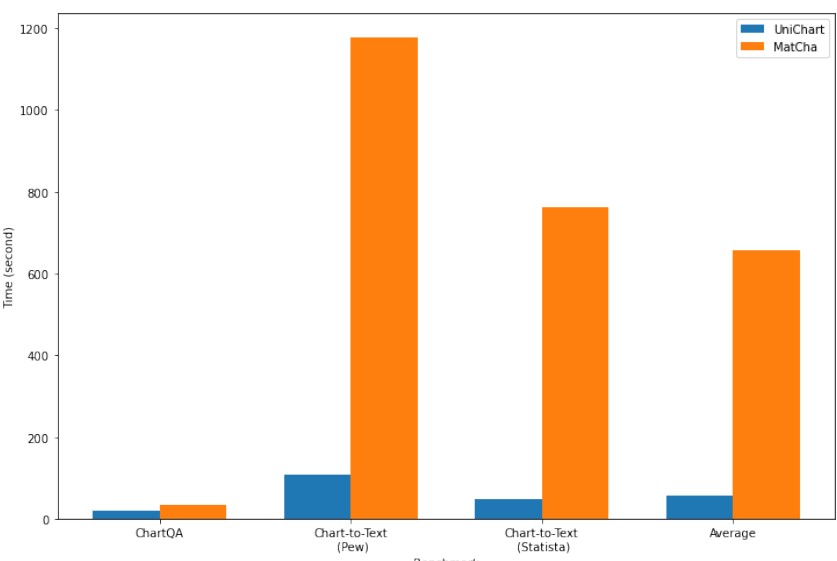

Figure 4: Average inference time for 10 random samples from three major benchmarks in chart understanding domain for UniChart and MatCha models

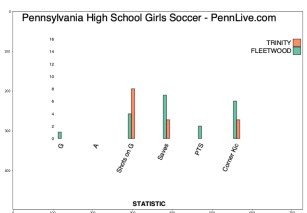

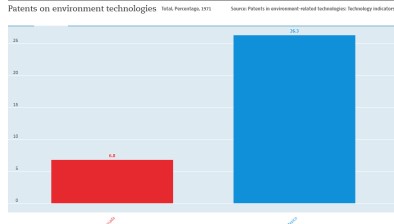

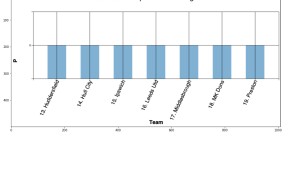

The bar chart compares the performance of two high school girls soccer teams, Fleetwood and Trinity, in a game. Both teams had a goal difference of 1.0, with Fleetwood scoring 1.0 goal and Trinity scoring 0.0. Both teams had 0.0 in shots on goal, saves, points, and corner kicks. However, Fleetwood had 4.0 shots on goal, 7.0 saves, 2.0 points, and 6.0 corner kicks, while Trinity had 8.0 shots on goal, 3.0 saves, 2.0 points, and 3.0 corner kicks. Overall, the chart shows that both teams had similar performances in the game, with Fleetwood scoring 1.0 goal and Trinity scoring 0.0.

The bar chart compares the number of patents on environment technologies between Canada and Mexico. Canada has 6.8 patents on environment technologies, while Mexico has 26.3. This indicates that Mexico has more than double the number of patents on environment technologies compared to Canada. This suggests that Mexico is more active in the field of environment technologies.

The bar chart shows the results of the Vital Leeds poll in chronological order for seven teams: Huddersfield, Hull City, Ipswich, Leeds United, Middlesbrough, MK Dons, and Preston. All teams had a score of 0.0 in the poll, indicating that they had not been voted on yet. This suggests that the team was not yet considered to be a clear favorite in the poll, and that the results of the poll may be skewed by the outcome of the next few games.

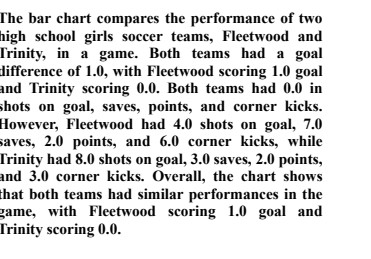

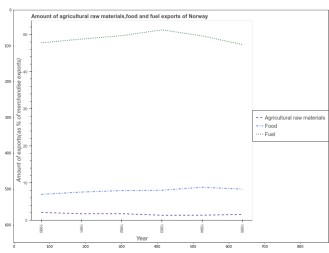

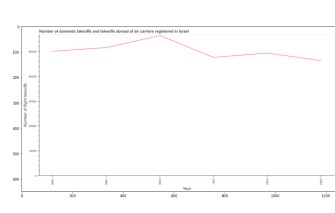

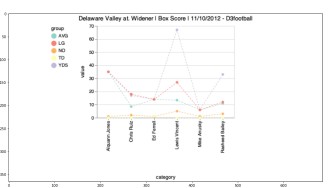

The line chart shows the amount of agricultural raw materials, food, and fuel exports of Norway from 1990 to 1995. Agricultural raw materials exports decreased from 2.18% in 1990 to 1.32% in 1993. Food exports increased from 6.94% in 1990 to 8.85% in 1994. Fuel exports increased from 47.78% in 1990 to 51.26% in 1993. Overall, the trend shows a decrease in the amount of agricultural raw materials, food, and fuel exports of Norway from 1990 to 1995.

The line chart displays the number of domestic takeoffs and takeoffs abroad of air carriers registered in Israel over a six-year period from 1998 to 2003. The number of flight takeoffs started at 50,100 in 1998, increased to 51,010 in 1999, then decreased to 56,427 in 2000, decreased further to 47,648 in 2001, increased to 49,404 in 2002, and decreased again to 46,334 in 2003. Therefore, the chart depicts a fluctuating trend of the number of flight takeoffs of air carriers registered in Israel during the six-year period, with an overall decreasing trend from 1998 to 2003.

The line chart shows the statistical performance of individual players from Delaware Valley and Widener in a football game on November 10, 2012. The categories measured are number of carries (NO), yards gained (YDS), average yards per carry (AVG), longest gain (LG), and touchdowns scored (TD). Among the players, Lewis Vincent had the most carries (5.09) and yards gained (64.09), with an impressive average of 13.09 yards per carry. Chris Ruiz also had a strong performance, with an average of 8.42 yards per carry and 0.42 touchdowns scored. Meanwhile, Mike Anusky had the fewest carries (1.42) and yards gained (6.09), but still managed to score a touchdown. Overall, the chart shows that the players from both teams had varying levels of performance in the game.

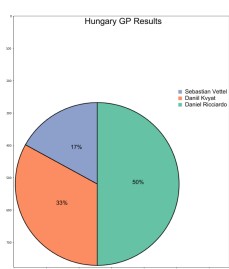

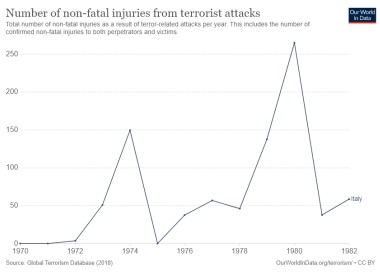

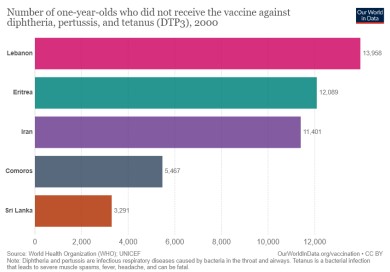

The pie chart shows the results of the Hungarian Grand Prix motor race in 2020, categorized by the winner of the race. The winner of the race was Sebastian Vettel, who accounted for 17% of the total results, followed by Daniil Kvyat at 33%, and Daniel Ricciardo at 50%. This chart indicates that the race was closely contested, with all three drivers finishing within a close margin of each other.

The line chart displays the number of non-fatal injuries from terrorist attacks over a nine-year period from 1970 to 1982. The x-axis shows the number of non-fatal injuries, while the y-axis shows the number of fatalities. The number of non-fatal injuries increased from 4.23 in 1970 to 271.09 in 1980, before decreasing to 42.18 in 1981. Overall, the chart shows an increasing trend in the number of non-fatal injuries from terrorist attacks during the nine-year period, with a slight decrease in 1978.

The bar chart shows the number of one-year-olds who did not receive the vaccine against diphtheria, pertussis, and tetanus (DTP3) in 2000 for five countries: Lebanon, Eritrea, Iran, Comoros, and Sri Lanka. Lebanon had the highest number of one-year-olds who did not receive the vaccine with 13,958 while Eritrea had 12,089 and Iran had 11,401. Comoros had 5,467 and Sri Lanka had 3,291. The data suggests that there is a significant variation in the number of one-year-olds who did not receive the vaccine against diphtheria, pertussis, and tetanus across these countries.

Figure 5: Examples of summaries generated by Flan-T5 XL model after fine-tuning.

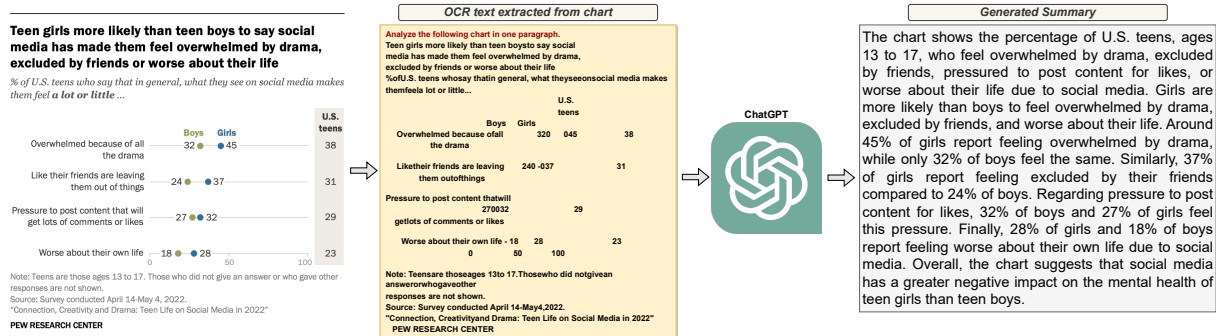

Figure 6: An example of the layout-preserved OCR-extracted text for a PewResearch chart image where the underlying data table is not available. The extracted text is then given to ChatGPT to generate a summary. ChatGPT can still extract and comprehend important information and insights from the layout-preserving text of the chart image.

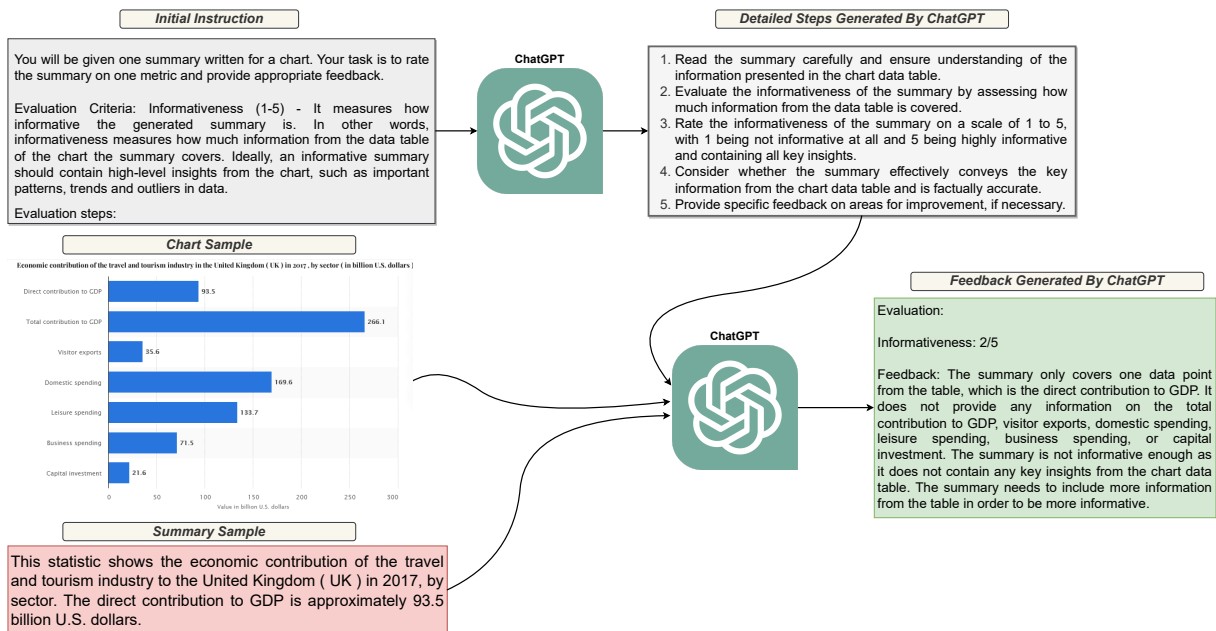

Figure 7: The pipeline designed for the ChatGPT Evaluation Experiment. First, we feed the task description followed by our desired criteria into ChatGPT in order to get detailed grading instructions. Then, the chart (underlying data table representation) and a sample summary are appended to the prompt which is fed again into ChatGPT to receive the feedback.

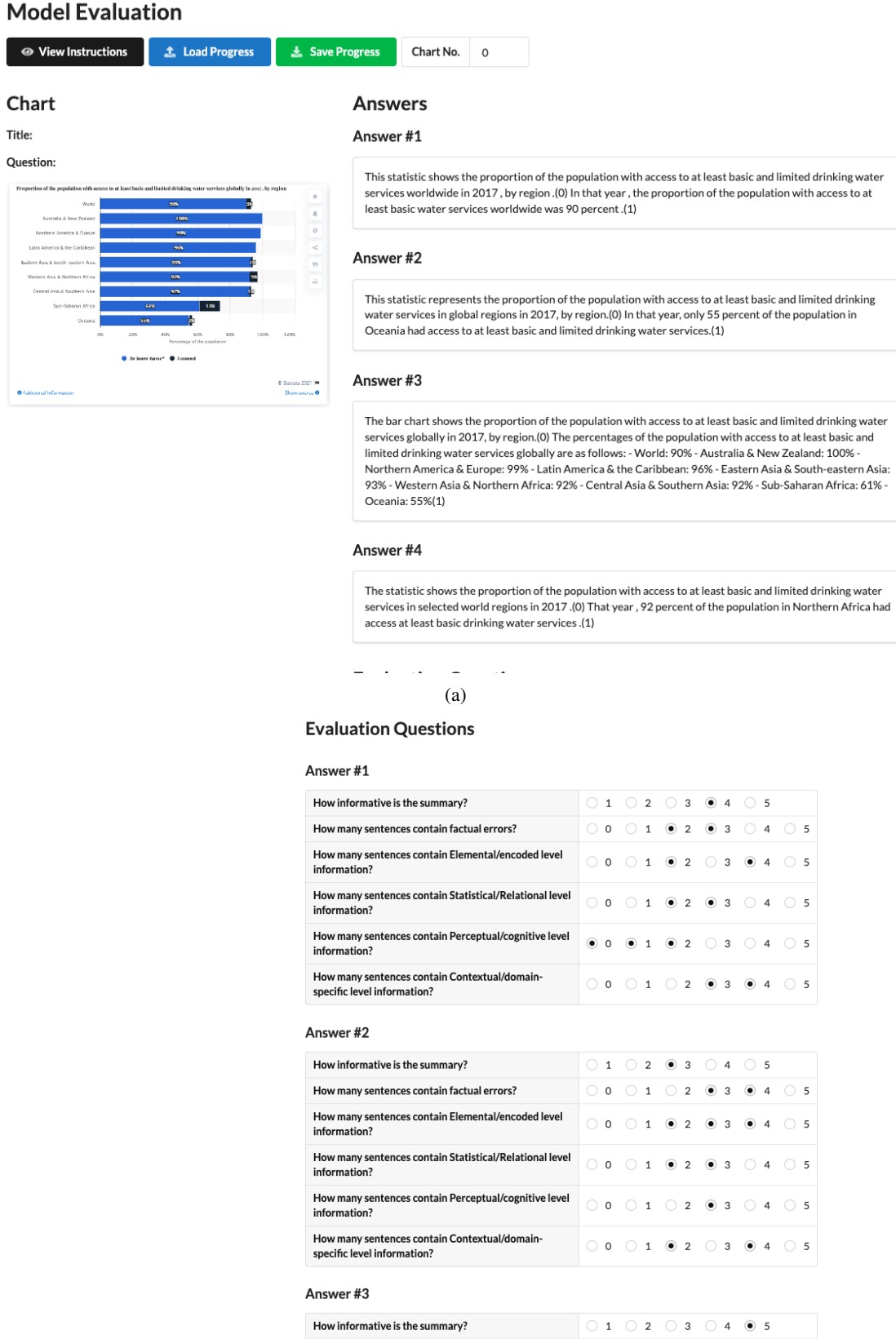

(a)

(b)

Figure 8: Human evaluation interface.

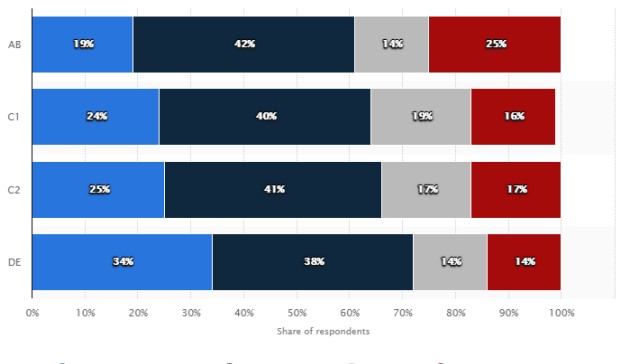

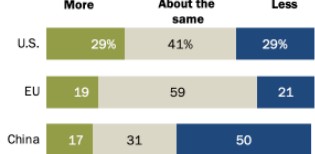

**Q1**: What is the largest value of red bar?
**Answer**: 25    *UniChart*: 25    **MatCha**: 40

**Q2**: What's the average of all the values in the green bars (round to one decimal)?
**Answer**: 21.6    *UniChart*: 0.4    **MatCha**:0.51

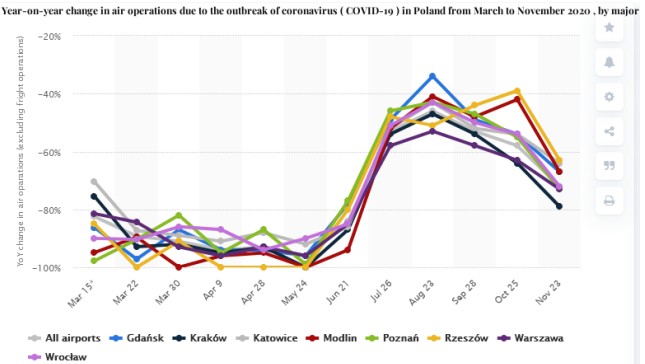

**Q3**: Summarize the chart.
**UniChart ZeroShot***: The line chart shows the year-on-year change in air operations due to the outbreak of COVID-19 in Poland from March to November 2020, broken down by major occupations. The occupations are measured in air operations (excluding Fight operations), in wholes (excluding Fight operations), and in wholes (excluding Fight operations).The occupations are measured in air operations (excluding Fight operations), in wholes (excluding Fight operations), and in wholes (excluding Fight operations). The occupations are measured in air operations (excluding Fight operations), in wholes (excluding Fight operations), and in wholes (excluding Fight operations). The occupations are measured in air operations (excluding Fight operations), in wholes (excluding Fight operations), ............................................................ revehivehivehivehivehivehivehivehivehivehivehivehivehivehivehivehivehiveh.

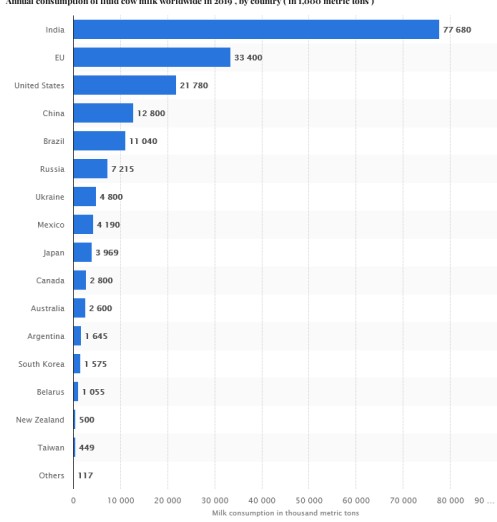

**Q4**: Summarize the chart.
**UniChart Finetuned**: This statistic illustrates the annual consumption of fluid cow milk worldwide in 2019, by country. In that year, India's fluid cow milk amounted to 77.7 million metric tons, making it the world's second largest consumer of milk.

Figure 9: Some challenging examples from the ChartQA and Chart-to-text benchmarks.

**Template-based Questions**

1) First bar from the top/left in the second group from the top/left
2) First bar from the bottom/right in the second group from the bottom/right
3) Second bar from the bottom/right in the first group from the bottom/right
4) Second bar from the right/bottom in the first group from the left/top
5) Topmost/Leftmost bar
6) Bottommost/Rightmost bar
7) Second bar from the top/left
8) Second bar from the right/bottom
9) Leftmost topmost bar
10) Leftmost bottommost bar
11) Rightmost topmost bar
12) Rightmost bottommost bar
13) Leftmost `<color>` data
14) Rightmost `<color>` data
15) Second from the left `<color>` data
16) Second from the right `<color>` data
17) Which legend represented by `<color>`?
18) What is the color of `<legend>`?
19) Which one is greater, `<x1>` or `<x2>`?
20) Divide the sum of largest and lowest values by `<n>`
21) When did line `<legend - label>` peak?
22) What is the difference between maximum and minimum of `<legend - label>`?
23) Sum pie segments above `<value>`
24) What is the sum of top three values?
25) What is the median/mode of `<legend - label>`?
26) What is the negative peak of `<legend - label>`?
27) What is the largest/smallest value of `<legend - label>`?
28) Which two x-axis labels of `<legend - label>` sums up to `<value>`?
29) What is the sum of the second highest and second lowest value of `<legend - label>`?
30) Which x-axis label is second highest for `<legend - label>`?
31) What is the sum of two middle values of `<legend - label>`?
32) Which two x-axis labels of `<legend - label>` have a difference of `<value>` ?
33) What is the average of `<legend - label>` from `<x - label - 1>` to `<x - label - 2>`?
34) What is the average of the highest and lowest value of `<legend - label - l>`?
35) What is the sum of the average of `<legend - label - 1>` and average of `<legend - label - 2>`?
36) What is the sum/difference of the maximum of `<legend - label - 1>` and minimum of `<legend - label - 2>`?
37) Which x-axis label has the maximum/minimum difference between `<legend - label - 1>` and minimum of `<legend - label - 2>`?
38) Which x-axis label witnessed the smallest value of `<legend - label>`?
39) Which label contains largest/smallest values across all labels?
40) Sum up the medians of all the data series in this chart
41) What is the average of all values above `<value>`?
42) What is the sum of the largest and smallest difference between `<legend - label - 1>` and `<legend - label - 2>`?
43) What is the maximum/minimum difference between `<legend - label - 1>` and `<legend - label - 2>`?
44) What is the ratio of the largest to the smallest pie segment?
45) What is the ratio of the two largest/smallest segments?
46) What is the difference between the leftmost and rightmost bars?
47) What is the sum of the bars in the second group from the left?
48) What is the sum of the bars in the first group from the right?
49) What is the ratio between the two leftmost bars?
50) What is the difference between the rightmost `<color - 1>` bar and leftmost `<color - 2>` bar?
51) What is the average of `<color>` bars values?
52) How many `<color>`bars are larger than `<N>` ?
53) What is the average of the bars in the second group from the right?
54) How many bars in the leftmost group have a value over `<N>`?
55) What does the `<color>` represent?
56) What is the median value of the `<color>` bars/line?
57) What is the average of the `<color - 1>` sum and `<color - 2>` sum?
58) What is the average of the `<color - 1>` median and `<color - 2>` median?
59) What is the least difference between the `<color - 1>` and `<color - 2>` bars/line?
60) What is the ratio between the leftmost and rightmost bar in the first group from the left?
61) What is the maximum value in the `<color>` bars/line?
62) What is the minimum value in the `<color>` bars/line?
63) What is the sum of `<color>` bars/line?
64) What is the difference between the maximum values of the two leftmost bar groups?
65) Sum of the first `<color - 1>` and last `<color - 2>` bars/line points
66) Difference between the two lowest `<color>` bars
67) Add largest and smallest `<color>` line/bar values and divide by 2
68) What is the value of `<color>` line/bars in `<x - axis - label>`?
69) Sum/Average of `<color - 1>` and `<color - 2>` values in `<x - axis - label>`?
70) Sum of highest points in `<color - 1>` and `<color - 2>` lines/bars
71) Which color has the highest/smallest values?
72) How many values are equal in `<color - 1>` line/bar?
73) Sum two rightmost values of `<color>` graph
74) Product of two smallest values in the graph
75) Sum of lowest and median values of `<color>` graph/bars
76) When did `<color>` line reached the peak?
77) What is the average of the rightmost three points of `<color>` line?
78) How many `<color>` data points are above `<value>`?
79) What's the ratio of the largest and the third/second-largest `<color>` bar?
80) Is the sum of lowest value of `<color - 1>` and `<color - 2>` bar greater than largest value of `<color - 3>` bar?
81) Is the median value of `<color - 1>` bars greater than the median value of `<color - 2>` bars?
82) Is the median of all the `<color - 1>` bars greater than the largest value of `<color - 2>` bar?
83) What's the product of `<color>` bars in India and Japan?
84) Is the sum of the two middle bars greater than the sum of top and bottom bars?
85) What's the ratio of the `<x - axis - label - 1>` `<color - 1>` bar and the `<x - axis - 2>` `<color - 2>` bar?
86) Is the total of all `<color - 1>` bars greater than the total of all `<color - 2>` bars?
87) Take the sum of the two smallest `<color - 1>` bars and smallest `<color - 2>` bars, deduct the smaller value from the larger value, what's the result?
88) What is the sum/average of two smallest/largest `<color>` bars?
89) What is the ratio of `<color - 1>` and `<color - 2>` segments?
90) What segment is represented by `<color>`?

Table 9: Numerical & Visual reasoning templates.