# OpenReview forum: "UniChart: A Universal Vision-language Pretrained Model for Chart Comprehension and Reasoning"
_EMNLP/2023/Conference — EMNLP 2023 Main_

### Official Review · Reviewer_hLdJ · 2023-08-05

**Soundness:** 3

**Excitement:**

4: Strong: This paper deepens the understanding of some phenomenon or lowers the barriers to an existing research direction.

**Missing References:**

ChartReader: A Unified Framework for Chart Derendering and Comprehension without Heuristic Rules. ICCV, 2023

**Paper Topic And Main Contributions:**

The paper introduces UniChart, a pre-trained model developed specifically for understanding and reasoning based on charts. This is an innovative area of study that addresses an important need in the AI field, as visual representations like charts are a fundamental way humans communicate information.

The main contributions of this paper are:

1. The development of UniChart, a pre-trained model designed for chart comprehension and reasoning. The model encodes related text, data, and visual elements of charts, and utilizes a text decoder for text generation based on the chart.

2. The creation of a large-scale pre-training dataset for charts. This dataset is versatile and extensive, covering a variety of topics and visual styles, and providing a rich resource for training models.

3. Through pre-training UniChart on this extensive chart corpus, and then fine-tuning it, the authors demonstrate that the model can achieve state-of-the-art performance on four downstream tasks. Notably, UniChart also demonstrates strong generalization capabilities, outperforming methods that lack chart-specific targets and rely on limited chart resources.

**Questions For The Authors:**

1. In Table 2, you've compared UniChart with ChatGPT. Could you please clarify how this comparison was conducted? Specifically, I'm interested in knowing whether images were used as input for ChatGPT. Further, I would like to know whether this comparison was fair and reasonable given the fact that ChatGPT is primarily a text-based model.

2. In the same table, it would be beneficial to include the number of parameters for each of the compared methods. This would provide a more detailed overview of the computational complexity involved with each model.

3. For the statistical analysis, could you elaborate on the significance and interpretation of the p-values you've reported in your results?

4. In regard to Table 1, I would like to understand whether all the data mentioned was used for pre-training. If there were any data filtering or preprocessing steps involved, please provide some information about them. Additionally, it would be helpful to know where this collected dataset can be accessed.

5. Lastly, details about the model training are not clearly outlined in the paper. Were the weights of the Chart image encoder and text decoder pre-trained on other datasets, or were they trained from scratch?

**Reasons To Accept:**

1. The authors have constructed a large-scale pre-training dataset for charts covering a wide range of themes and visual styles. This is a significant contribution as such a diverse and large dataset can potentially facilitate more research in this area.

2. The model demonstrated state-of-the-art performance on four downstream tasks, showcasing its effectiveness. Moreover, it showed strong generalization capabilities on unseen chart data, which suggests it has been well-trained and has broad applicability.

**Reasons To Reject:**

1. Lack of Model Comparison Details: The authors compare their model, UniChart, with ChatGPT in Table 2 without providing sufficient details about how the comparison was conducted. Given that ChatGPT is not designed for visual input, it raises concerns about the fairness and validity of this comparison.

2. Incomplete Presentation of Models: The authors did not include important information, such as the number of parameters for each model, in their comparison. This omission limits the reader's understanding of the complexity and computational demands of the proposed model compared to others.

3. Ambiguity in Statistical Interpretation: The authors used p-values in their analysis without properly explaining their significance or interpretation, causing confusion for readers unfamiliar with statistical testing.

4. Insufficient Information on Data Preprocessing: The paper lacks a clear description of the data preprocessing steps and whether all data in Table 1 was used for pre-training. This omission hinders the reproducibility of the results.

5. Unclear Training Details: The authors did not specify whether the Chart image encoder and text decoder were trained from scratch or used pre-trained weights from other datasets. This lack of information further limits the replicability of the research and the full understanding of the model's capabilities.

**Reproducibility:**

3: Could reproduce the results with some difficulty. The settings of parameters are underspecified or subjectively determined; the training/evaluation data are not widely available.

**Reviewer Confidence:**

4: Quite sure. I tried to check the important points carefully. It's unlikely, though conceivable, that I missed something that should affect my ratings.

---

> ### Author Rebuttal · Authors · 2023-08-29
>
> # Response to concerns in the Reasons To Reject and the Reviewer Questions
>
> 1. ChatGPT (Reason 1 and Question 1)
>     * **Response:** We did not compare UniChart results to ChatGPT. In fact, there is even no mentioning of ChatGPT in Table 2. We used ChatGPT only in evaluating the summaries generated by UniChart and other baselines. To do this, we fed the underlying data table of the chart in text format beside the generated summary to ChatGPT and asked to rate the summary. The detailed pipeline can be found in Figure 7 in the paper.
>
> 2. Number of parameters (Reason 2 and Question 2)
>     * **Response:** We will elaborate on the number of parameters and pretraining data for each model in Table 2 in the paper.
>
> 3. Statistical Interpretation in Table 3 (Reason 3 and Question 3)
>     * **Response:** We did a statistical test (p-value) for ratings from humans and ChatGPT, with the null hypothesis that the ratings are two independent samples. The p-values in each row in Table 3 demonstrate that it is very infrequent that two rating samples are independent based on the observed ratings. We will elaborate more on this point in the final version of the paper.
>
> 4. Dataset Preprocessing (Reason 4 and Question 4)
>     * **Response:** The numbers in Table 1 represent the information after preprocessing the data. During the data preprocessing, we excluded validation and test sets from all the sources that were shared among downstream tasks like Statista and Pew.  Our model was pretrained on all the examples mentioned in Table 1. The dataset will be published in the project repository after the anonymity period.
>
> 5. Model Training Details (Reason 5 and Question 5)
>     * **Response:** As mentioned in the A.4 section in Appendix (lines 1289-1291), we initialized the weights from the base Donut weights. Given the importance of this information, we will move this information to the main text from Appendix.

---

### Official Review · Reviewer_8Lrx · 2023-08-05

**Typos Grammar Style And Presentation Improvements:** No
**Soundness:** 3

**Excitement:**

3: Ambivalent: It has merits (e.g., it reports state-of-the-art results, the idea is nice), but there are key weaknesses (e.g., it describes incremental work), and it can significantly benefit from another round of revision. However, I won't object to accepting it if my co-reviewers champion it.

**Missing References:**

No

**Paper Topic And Main Contributions:**

This paper aims to address the problem that current approaches to chart analysis ignore explicit modeling of chart structure, and to this end, proposes a generalized visual language pretraining model for chart comprehension and reasoning, UniChart ,which utilizes a chart-based text decoder to encode text, data, and visual elements. The authors propose pre-training tasks for charts, including low-level element extraction and high-level comprehension. Pre-training UniChart on a large scale and then fine-tuning it achieved state-of-art performance in all four tasks.

**Questions For The Authors:**

1.	The authors have said that text-davinci-003 can generate “coherent and relevant text”. However, after given the generated text by text-davinci-003, LLM can also do all the tasks (like question answer, summarization and so on). What is the advantage to use UniChart compared to directly using LLM like Flan-T5 XL and other LLMs? Especially, can the UniChart perform better than “text-davinci-003” (by using text-davinci-003 to generate text, and then answer questions and summarize over these generated texts).

2.	There are also many data-to-text methods. If the Chat can be converted to a data table (by OCR and other operations), what is the advantage of using UniChart compared to converting the Chat to the data table, and then using the data-to-text method to conduct these tasks?


**Reasons To Accept:**

1．	The paper highlights the limitations of existing methods in chart-based data analysis tasks. These methods often rely on pretraining from language or vision-language tasks, but overlook the explicit modeling of chart structures.

2．	The paper employs an extensive dataset for pretraining, with the authors dedicating significant efforts to the collection and processing of the dataset.


**Reasons To Reject:**

1. There are certain descriptions that lack precision. For instance, in lines 226-229, there is a statement that reads: "Our goal was to train the model based on real-world data, thus, we did not consider the ones that are generated from synthetic data." However, in the subsequent "Data Augmentation" section, the authors proceed to construct charts based on data tables, which seems contradictory.  Therefore, the intent that the authors wanted to convey in lines 226-229 should be expressed in a clearer and more accurate manner, so as to avoid inconsistency with the subsequent content.

2. Ablation experiments were only done on the ChartQA dataset, which I think is somewhat inadequate.

3. I have concerns about the application of UniChart, comparing the UniChart to LLMs and the data-to-text method. Please see my question as below.


**Reproducibility:**

3: Could reproduce the results with some difficulty. The settings of parameters are underspecified or subjectively determined; the training/evaluation data are not widely available.

**Reviewer Confidence:**

4: Quite sure. I tried to check the important points carefully. It's unlikely, though conceivable, that I missed something that should affect my ratings.

---

> ### Author Rebuttal · Authors · 2023-08-29
>
> # Response to concerns in the Reasons To Reject and the Reviewer Questions
>
> 1. Real-world data in Data Augmentation (Reason 1)
>
>     * **Response:** Unlike corpus from DVQA/FigureQA which randomly generates the data table values, we only used real-world data tables during training. Our data augmentation pipeline is based on the data tables we extracted from WDC corpus which is a source of real world data tables available on the internet.
>
> 2. Ablations experiments on ChartQA (Reason 2)
>
>     * **Response:** As we discussed in the paper (lines 551-556), automatic evaluation metrics like BLEU score for text generation benchmarks like Chart-to-Text and OpenCQA are not very informative. In contrast, the results of ChartQA are more interpretable as they are based on simple relaxed accuracy measures. Therefore we decided to do ablation studies only for ChartQA which can demonstrate the performance of different setups clearly.
>
> 3. UniChart vs LLMs and data-to-text models (Reason 3, Question 1, Question 2)
>
>     * **Response:** First, addressing questions regarding a chart requires more information from the chart than just extracted OCR text. For example, in the ChartQA benchmark, a common type of user queries are those that refer to visual elements of a chart (e.g., what is the least difference between light blue bar and dark blue bar?) which cannot be answered using OCR text only. It is already confirmed in the ChartQA paper that models that rely on the data tables only (e.g., TaPas and T5) exhibit lower performance than their multimodal counterparts (e.g., VisionTapas and VL-T5) which also take the chart image into consideration. Moreover, UniChart is an end-to-end reproducible, transparent, as well as memory and time-efficient solution for chart comprehension compared to models like text-davinci-003 which are not publicly available and have larger parameter counts.
>
> ## Reproducability
>    * **Response:** Due to the anonymity requirement, we removed our github repo link from the paper (lines 125-126). After the anonymity period is over, we will provide the github repo link that contains all the data and checkpoints.

---

### Official Review · Reviewer_bdEC · 2023-08-09

**Soundness:** 4

**Excitement:**

4: Strong: This paper deepens the understanding of some phenomenon or lowers the barriers to an existing research direction.

**Missing References:**

No major missing references that I am aware of.

**Paper Topic And Main Contributions:**

This paper proposes a new chart-specific pretraining dataset, as well as a new pretrained model for chart comprehension and reasoning. The dataset is derived from several sources: they extract the actual svg charts from various web sources, augment the corpus by generating new charts from data tables with the D3 framework, and obtain image-only charts which they furnish with textual content by applying OCR. They train a new model, UniChart, on this corpus, and show that it outperforms existing models on several evaluation datasets.

**Questions For The Authors:**

- Can you provide ablation experiments that would allow us to understand the effect of the pretraining corpus vs. the proposed UniChart model architecture? Which contribution is more important for better performance, and by how much?
- What are the parameter counts of each of the models in Table 2? It would be great if the parameters and pretraining data used could be listed here for ease of comparison.
- Did you perform de-duplication of the evaluation datasets used in Table 2 against the pretraining corpus? This is important to ensure that none of the charts used in validation leaked into the pretraining dataset.



**Reasons To Accept:**

- The paper prepares and releases a new large pretraining corpus for chart data. This data is derived from real world sources, as well as generated from real-world data tables. It will be a valuable dataset for the community.
- The paper proposes a new model and pretrains it on the chart dataset, showing that it outperforms prior work on several tasks. Evaluations are comprehensive, spanning automated metrics (BLEU, RA), human evaluations, and LLM evaluations.
- The error analysis in the paper highlights important drawbacks of the existing model, and suggests ways to improve them in future work.
- The evaluation and analysis provided in the main paper and the appendix is very comprehensive.


**Reasons To Reject:**

- There is an important ablation missing in this paper to compare architecture against pretraining data. The paper proposes both a new dataset, as well as a new model architecture. It’s unclear whether the improved results are due to a better model architecture, or cleaner pretraining data (I suspect the latter). How do prior models do when they are pretrained on the same corpus? In particular, I would like to see results on MatCha or Pix2Struct if it was pretrained (or finetuned) on the same dataset as UniChart.
- The ChatGPT evaluation is a bit questionable, given the non-determinism and opaqueness of ChatGPT, and its limitations as a model for providing real (or integer) valued ratings (the paper seems to get ChatGPT to generate ratings from 1-5). One of the cited papers, Luo et al. (2023) also highlights the limitations of such an approach. A better evaluation might be to give ChatGPT inputs from two different models, and ask it to rank them instead. This type of evaluation is more robust than generating score outputs. However, it’s good that the results generally correlate with human scores, so this is not a big issue. I would encourage the authors to de-emphasize the ChatGPT evaluations (maybe even move it to the appendix), given that we don’t know the model details, whether the model has changed since the evaluations were run, and other implementation details.


**Reproducibility:**

4: Could mostly reproduce the results, but there may be some variation because of sample variance or minor variations in their interpretation of the protocol or method.

**Reviewer Confidence:**

4: Quite sure. I tried to check the important points carefully. It's unlikely, though conceivable, that I missed something that should affect my ratings.

**Typos Grammar Style And Presentation Improvements:**

The paper is generally well written and easy to follow. The font size of Table 1 could be increased as it is difficult to read when printed.

---

> ### Author Rebuttal · Authors · 2023-08-29
>
> # Response to concerns in the Reasons To Reject and the Reviewer Questions
> 1.  MatCha/Pix2Struct ablation studies (Reason 1 and Question 1)
>
>     *  **Response:** We chose Donut architecture as the backbone model since it was much more efficient in terms of computational complexity during both training and inference time (as described in lines 654-668), especially for the high-resolution chart images. Our pretraining computational resources were limited (4xA100 80GB) compared to the computational resources needed for Pix2Struct/MatCha. It is also worth noting that the original MatCha model from Google was pretrained using 64 GCP-TPUv3 which is infeasible in an academic setup. Moreover, based on the results reported in the Pix2struct paper, fine-tuned Donut exhibits lower performance compared to Pix2struct on ChartQA benchmark, which suggests the effect of our pretraining corpus  in improving the results rather than the pretrained backbone model.
>
> 2. ChatGPT evaluation (Reason 2)
>
>     * **Response:** We opted for the rating strategy (Table 3), mainly because it was less expensive and had less latency compared to pairwise comparisons (more API calls). However, to address the reviewer’s comment, we did an additional analysis using pairwise comparisons executed by ChatGPT on the same 150 random samples used in Table 3 (see below). The findings from this analysis align with our initial conclusions regarding the performance of various models. The numbers below represent the proportion of instances where ChatGPT favored one model over another within the pairwise comparison framework.
>
>
>         * ZeroShot vs Finetuned: (ZeroShot wins: 83.8%, Finetuned  wins: 7.7%, Tie: 7.7%)
>
>         * ZeroShot vs MatCha: (ZeroShot wins: 89.5%, MatCha wins: 6.9%, 3.4% Tie)
>
>         * Finetuned vs MatCha: (Finetuned wins: 48.5%, MatCha wins: 35%, 16.4% Tie)
>
>         * Finetuned vs Gold: (Finetuned wins: 28.3%, Gold wins: 53.9%, 17.7 Tie)
>
>         * Gold vs MatCha: (Gold wins: 62.3%, MatCha wins: 23.9%, 15.7% Tie)
>
>         * Gold vs ZeroShot: (Gold wins: 27.4%, ZeroShot wins: 62.6%, 9.8% Tie)
>
>     * Based on the reviewer's suggestion, we will also de-emphasize ChatGPT evaluation and move it into the appendix.
>
> 3. Parameters Count for the models in Table 2 in the paper (Question 2)
>
>    * **Response:** We will elaborate the number of parameters and pretraining data for each model in Table 2 in the paper.
> 4. De-duplication of the evaluation datasets (Question 3)
>     * **Response:**  Yes, we excluded validation and test sets from all the sources that were shared among downstream tasks like Statista and Pew. We will clarify this further in the paper.

---

### Meta-Review · Area_Chair_tBV8 · 2023-09-18

**Recommendation:** 5

**Metareview:**

All reviewers felt generally positively about both the soundness and excitement of the paper, describing the proposed pre-training dataset as "real world... valuable", "extensive", and "diverse" with the evaluation of the new proposed model as "comprehensive", showing its effectiveness on several downstream datasets, and appreciated the analysis-based suggestions for future work on the dataset and task. Some smaller reviewer concerns about presentation/clarity of details, and ablations, were adequately addressed in the author response.

---

### Decision · Program_Chairs · 2023-10-07

**Decision:**

Accept-Main

**Comment:**

All reviewers felt generally positively about both the soundness and excitement of the paper, describing the proposed pre-training dataset as "real world... valuable", "extensive", and "diverse" with the evaluation of the new proposed model as "comprehensive", showing its effectiveness on several downstream datasets, and appreciated the analysis-based suggestions for future work on the dataset and task. Some smaller reviewer concerns about presentation/clarity of details, and ablations, were adequately addressed in the author response.